# GEOMETRY-GROUNDED FLOW MATCHING ON COMPACT MANIFOLDS

**Ali Baheri**
Department of Mechanical Engineering
Rochester Institute of Technology
Rochester, NY 14623, USA
akbeme@rit.edu

## ABSTRACT

Riemannian Flow Matching extends Flow Matching generative modeling to data that lives on curved spaces such as spheres and tori by learning a time-dependent vector field and generating samples through ordinary differential equation integration. This paper provides an end-to-end theoretical guaranty for the standard Riemannian Flow Matching pipeline on compact manifolds. Our analysis separates three sources of error: the statistical error from learning the conditional-mean velocity field produced by conditional flow matching, the approximation and optimization error arising from the chosen function class and empirical risk minimization, and the discretization error introduced by the numerical ODE solver. A key technical contribution is a flow-to-distribution stability result that is robust to geometry: curvature and injectivity radius influence only constants under standard boundedness and Lipschitz regularity conditions. Under metric-entropy assumptions, the learning rate is governed by the intrinsic manifold dimension rather than any ambient embedding dimension. Experiments on the circle, the sphere, and the two-torus support the predicted scaling behavior.

## 1   INTRODUCTION

Many datasets and latent representations are inherently *non-Euclidean*: directions lie on spheres, periodic variables live on tori, and rigid-body orientations belong to Lie groups. In such settings, naively applying Euclidean generative models can be statistically and algorithmically mismatched; samples may violate constraints, distances may be distorted, and learning rates may depend on an ambient embedding dimension rather than the intrinsic geometry. This has motivated a growing literature on *Riemannian* generative modeling, including manifold diffusion models (De Bortoli et al., 2022) and, more recently, *Riemannian Flow Matching* (RFM) (Chen & Lipman, 2024), which learns a time-dependent vector field on the manifold and generates samples by integrating an ODE.

Flow matching is appealing in practice because training reduces to a regression-style objective for a velocity field, while sampling is deterministic via ODE integration. In Euclidean space, several recent works have begun to clarify the statistical behavior of flow matching and related velocity-learning pipelines (Benton et al., 2024; Fukumizu et al., 2024). However, transferring such guaranties to curved spaces raises additional difficulties. First, the stability of flows depends on geometric control (curvature, injectivity radius) and the regularity of the learned vector field. Second, the end-to-end error of a generative pipeline mixes distinct sources: regression/statistical error from learning a vector field, approximation error from the function class (e.g., neural parameterization), and numerical error from discretizing the ODE. A clean account of how these ingredients combine on manifolds is still limited.

This paper provides an explicit upper-bound analysis for the standard RFM pipeline on compact Riemannian manifolds. Our starting point is the conditional-flow-matching (CFM) training construction: it induces supervised regression samples $(T, X_T, Y_T)$, where $Y_T$ is a conditional velocity label at time $T$ and state $X_T$. The squared-loss population minimizer is the conditional mean vector field $v^*(t,x) = \mathbb{E}[Y_T \mid T = t, X_T = x]$. We study estimators that learn an approximation $\hat{v}$

to this regression target from $n$ samples and then output the pushforward distribution obtained by numerically integrating the ODE $\dot{x}_t = \hat{v}(t, x_t)$ from a fixed base measure.

**Our contributions.** Under bounded-geometry assumptions on the manifold and regularity/complexity conditions on the regression target and hypothesis class, we establish an end-to-end Wasserstein error bound for RFM that cleanly separates the main error sources.

- **Regression-to-transport error decomposition.** We formalize the two-step structure of RFM: a nonparametric regression problem for learning $v^\dagger$ from CFM samples, followed by transport through an ODE flow map. This yields an end-to-end bound on $W_2(\hat{\mu}_n, \mu^*)$ consisting of (i) a regression/statistical term, (ii) the approximation error of the hypothesis class, and (iii) discretization error from the numerical ODE solver (Corollary 1).
- **Curvature-stable flow-to-measure stability.** We provide a flow stability inequality on manifolds under uniform boundedness and spatial Lipschitz regularity of the vector fields. Curvature and injectivity radius enter through constants via comparison-geometry control, while the statistical exponent is governed by the regression rate (Theorem 2).
- **Intrinsic-dimension scaling and empirical study.** Specializing the regression analysis to metric-entropy conditions yields a conservative rate exponent $\alpha = \beta/(2\beta + d)$ (up to logarithmic factors) that depends on the intrinsic dimension $d$ rather than the ambient dimension (Theorem 1). We complement the theory with numerical experiments on $S^1$, $S^2$, and $T^2$, which exhibit clear power-law decay and intrinsic-dimension dependence consistent with the predicted scaling regime (Section 4).

**Organization.** Section 3 states the main theoretical results: a regression rate under metric-entropy assumptions, a flow-to-measure stability bound on manifolds, and an end-to-end Wasserstein guaranty for the full RFM pipeline. Section 4 reports numerical experiments on canonical manifolds and discusses how the observed scaling compares to the conservative theoretical prediction. Proofs and additional technical details are deferred to the appendix.

## 2  RELATED WORK

**Flow-based generative modeling and flow matching.** Continuous normalizing flows (CNFs) and neural ODE formulations provide a general ODE-based route to generative modeling Chen et al. (2018); Grathwohl et al. (2019). Flow matching (FM) reframes CNF training as supervised regression of a time-dependent velocity field along prescribed conditional probability paths, enabling scalable simulation-free training Lipman et al. (2022). Closely related ODE-first training paradigms include stochastic interpolants Albergo & Vanden-Eijnden (2023) and rectified flow Liu et al. (2023), which also emphasize quadratic objectives for learning transport dynamics and fast deterministic sampling. Beyond standard generative settings, structured variants of conditional flow matching have been proposed to encode inductive biases such as conservative–dissipative decompositions for learning physical dynamics Baheri & Lindemann (2025).

**Generative modeling on manifolds.** A growing literature studies generative modeling when data lie on curved spaces. Riemannian score-based/diffusion approaches extend diffusion models by replacing Euclidean SDE/score machinery with manifold counterparts De Bortoli et al. (2022); Huang et al. (2022). Riemannian Flow Matching (RFM) adapts FM to general geometries by learning a manifold vector field and sampling by integrating the induced ODE, avoiding expensive simulation and providing practical scalability on nontrivial geometries Chen & Lipman (2024). Related geometry-aware learning problems on manifolds also arise outside generation; for instance, conformal prediction methods have been adapted to use geodesic nonconformity scores to provide calibrated uncertainty regions in manifold-valued regression Amiri Shahbazi & Baheri (2026).

**Theory and end-to-end error bounds.** Recent theory for Euclidean FM/ODE-based generative pipelines has begun to quantify how velocity-learning error translates into distributional error, including deterministic-sampling error bounds Benton et al. (2024) and near-minimax convergence guaranties under suitable conditions Fukumizu et al. (2024). Our contribution complements this line by focusing on *manifold* pipelines: we provide an explicit end-to-end Wasserstein bound that (i)

treats CFM-induced learning as a nonparametric regression problem, (ii) provides a flow-to-measure stability inequality on compact manifolds where curvature enters primarily through constants under uniform Lipschitz regularity, and (iii) cleanly separates statistical, approximation, and numerical discretization effects within the standard RFM workflow. More broadly, Wasserstein/optimal-transport geometry has been used to study ambiguity and identifiability questions in other inference settings (e.g., inverse RL), offering perspectives on distributional geometry and uncertainty Baheri (2023).

## 3 MAIN RESULTS

This section provides end-to-end *upper bounds* for the standard Riemannian flow-matching pipeline on a compact Riemannian manifold: (i) construct i.i.d. regression data from conditional flow matching (CFM), (ii) learn a time-dependent vector field by squared-loss regression over a hypothesis class, and (iii) generate samples by numerically integrating the induced ODE and pushing forward a fixed base distribution. We include an explicit misspecification term, state stability with a well-defined trajectory norm, and assume a bounded "schedule" to avoid the canonical $t \to 1$ singularity of geodesic CFM velocities.

### 3.1 SETTING AND I.I.D. REGRESSION DATA FROM CONDITIONAL FLOW MATCHING

Let $(\mathcal{M}, g)$ be a connected, compact $d$-dimensional Riemannian manifold with geodesic distance $d_{\mathcal{M}}$. Fix a base distribution $\mu_0$ on $\mathcal{M}$. We observe i.i.d. samples $X_1, \ldots, X_n \sim \mu^*$.

**CFM regression dataset (explicit i.i.d. sampling).** Conditional flow matching produces supervised tuples by combining each data sample with independent auxiliary randomness. For each $i = 1, \ldots, n$:

1. sample $T_i \sim \mathrm{Unif}[0, 1]$ and $Z_i \sim \mu_0$, independently of everything else;

2. construct an interpolated state $X_{T_i}^{(i)} := \mathrm{Interp}(T_i; Z_i, X_i) \in \mathcal{M}$ (e.g., the minimizing geodesic interpolant when well-defined);

3. compute a tangent-space label

$$Y_i := U_\sigma(T_i, X_{T_i}^{(i)} \mid X_i) \in T_{X_{T_i}^{(i)}} \mathcal{M}, \tag{1}$$

where $U_\sigma$ is the chosen conditional-velocity rule with bounded "schedule" (Assumption 2).

By construction, $\{(T_i, X_{T_i}^{(i)}, Y_i)\}_{i=1}^n$ are i.i.d. across $i$.

**Population regression risk and target.** Let $(T, X_T, Y)$ denote a generic draw with the same law as $(T_i, X_{T_i}^{(i)}, Y_i)$. For a measurable field $v : [0, 1] \times \mathcal{M} \to T\mathcal{M}$, define the population risk

$$\mathcal{R}(v) := \mathbb{E}\big[\|v(T, X_T) - Y\|_{g_{X_T}}^2\big].$$

As in squared-loss regression, the (a.e.) minimizer is the conditional mean

$$v^\dagger(t, x) := \mathbb{E}[Y \mid T = t, X_T = x], \tag{2}$$

defined $p_t$-a.e., where $p_t$ is the density of $X_T$ given $T = t$ with respect to $\mathrm{vol}_g$.

**Learning rule (explicit).** Given the i.i.d. regression dataset, define the empirical risk

$$\hat{\mathcal{R}}_n(v) := \frac{1}{n} \sum_{i=1}^n \big\|v(T_i, X_{T_i}^{(i)}) - Y_i\big\|_{g_{X_{T_i}^{(i)}}}^2. \tag{3}$$

We fit an $\eta$-approximate ERM over a hypothesis class $\mathcal{F}_n$:

$$\hat{v} \in \mathcal{F}_n, \qquad \hat{\mathcal{R}}_n(\hat{v}) \le \inf_{v \in \mathcal{F}_n} \hat{\mathcal{R}}_n(v) + \eta.$$

**Flow-based estimator and solver.** Given a learned field $v$, let $\Phi_t^v$ be the time-$t$ flow map of the ODE $\dot{x}_t = v(t, x_t)$ on $\mathcal{M}$ (uniquely defined on compact manifolds under spatial Lipschitz regularity). A $K$-step numerical solver induces an approximate time-one map $\Phi_{1,K}^v$. The generative estimator is the pushforward

$$\hat{\mu}_n := (\Phi_{1,K}^{\hat{v}})_{\#}\mu_0,$$

and we measure error using the Wasserstein distance $W_2$ induced by $d_{\mathcal{M}}$.

## 3.2 Assumptions

**Assumption 1** (Bounded geometry). *$(\mathcal{M}, g)$ is compact with sectional curvature bounded by $|K| \leq \kappa$ and injectivity radius $\mathrm{inj}(\mathcal{M}) \geq r_0 > 0$.*

**Assumption 2** (Bounded schedule / no singular velocities). *The label rule $U_\sigma$ in equation 1 is measurable and satisfies $\mathbb{E}[\|Y\|_{g_{X_T}}^2] \leq M_2 < \infty$. A sufficient condition (matching common practice) is that $U_\sigma$ is defined using a bounded schedule $s_\sigma(t) \geq s_{\min} > 0$, e.g.*

$$U_\sigma(t, x \mid x_1) = \frac{\log_x(x_1)}{s_\sigma(t)}, \qquad s_\sigma(t) = 1 - t + \sigma \ \ (\sigma > 0),$$

*which avoids the canonical $(1-t)^{-1}$ singularity as $t \to 1$.*

**Assumption 3** (Training marginal regularity). *For $T \sim \mathrm{Unif}[0, 1]$, the conditional density $p_t$ of $X_T$ given $T = t$ satisfies $0 < p_{\min} \leq p_t(x) \leq p_{\max} < \infty$ for all $(t, x) \in [0, 1] \times \mathcal{M}$.*

**Assumption 4** (Flow regularity of the regression target and hypothesis class). *The regression target $v^\dagger$ is bounded and spatially Lipschitz:*

$$\sup_{t,x} \|v^\dagger(t, x)\|_{g_x} \leq B, \qquad \|v^\dagger(t, \cdot)\|_{\mathrm{Lip}} \leq L \ \text{for all } t \in [0, 1],$$

*and $\mathcal{F}_n$ is a class of vector fields with the same bounds: for all $v \in \mathcal{F}_n$, $\sup_{t,x} \|v(t, x)\| \leq B$ and $\|v(t, \cdot)\|_{\mathrm{Lip}} \leq L$.*

**Assumption 5** (Complexity and approximation). *Let $\|\cdot\|_\infty$ denote the supremum over $(t, x)$ of the pointwise tangent norm. There exists $\beta > 0$ such that for all $\epsilon \in (0, 1)$,*

$$\log \mathcal{N}(\mathcal{F}_n, \epsilon, \|\cdot\|_\infty) \leq C_{\mathrm{ent}} \, \epsilon^{-d/\beta} \, \mathrm{polylog}(1/\epsilon).$$

*Define the approximation error*

$$\mathcal{E}_{\mathrm{approx}}(\mathcal{F}_n) := \inf_{v \in \mathcal{F}_n} \left( \mathbb{E}\|v(T, X_T) - v^\dagger(T, X_T)\|_{g_{X_T}}^2 \right)^{1/2}.$$

**Remark 1** (Controlling time dependence). *Assumption 5 is stated in the supremum norm over $(t, x)$. It is satisfied by common parameterizations where the nonparametric complexity is primarily carried by $x$, and the $t$-dependence is represented with controlled capacity. For instance, one may use $v(t, x) = \sum_{k=1}^{K_t} \phi_k(t) f_k(x)$ with fixed $K_t$ and bounded basis functions $\phi_k$, while the spatial components $f_k$ range over a class whose entropy scales as $\epsilon^{-d/\beta}$.*

**Assumption 6** (Solver error). *There exists $\mathcal{E}_{\mathrm{disc}}(K)$ such that for any $v \in \mathcal{F}_n \cup \{v^\dagger\}$,*

$$\left( \mathbb{E}_{X_0 \sim \mu_0}\left[ d_{\mathcal{M}}(\Phi_1^v(X_0), \Phi_{1,K}^v(X_0))^2 \right] \right)^{1/2} \leq \mathcal{E}_{\mathrm{disc}}(K).$$

**Assumption 7** (Reference path consistency). *Let $\mu_t^\dagger := (\Phi_t^{v^\dagger})_{\#}\mu_0$. The training marginals coincide with this reference path: for each $t \in [0, 1]$, $p_t$ is the density of $\mu_t^\dagger$ with respect to $\mathrm{vol}_g$.*

**Remark 2.** *Assumption 7 links the CFM-induced regression target $v^\dagger$ to a reference flow path from $\mu_0$. It is the condition needed to convert a regression error measured under $p_t$ into a transport error along the reference flow.*

### 3.3 TWO-STEP GUARANTEES: REGRESSION ⇒ TRANSPORT

**Theorem 1** (Velocity regression under metric entropy). *Assume 3 and 5. Let $\hat{v}$ be an $\eta$-approximate ERM over $\mathcal{F}_n$ for equation 3. Then for any $\delta \in (0,1)$, with probability at least $1 - \delta$,*

$$\left( \mathbb{E}\|\hat{v}(T, X_T) - v^\dagger(T, X_T)\|_{g_{X_T}}^2 \right)^{1/2} \leq C_{\text{reg}}\, n^{-\frac{\beta}{2\beta+d}} \left( \log \frac{n}{\delta} \right)^\gamma + \mathcal{E}_{\text{approx}}(\mathcal{F}_n) + \eta^{1/2},$$

*where $C_{\text{reg}}, \gamma$ depends on $(p_{\min}, p_{\max}, C_{\text{ent}})$.*

**Theorem 2** (Flow-to-measure stability in $W_2$ (trajectory norm)). *Assume 1 and 4. For any two vector fields $v, w$ satisfying the common bounds $(B, L)$, define $\mu_t^v := (\Phi_t^v)_\# \mu_0$. Then*

$$W_2\big((\Phi_1^v)_\# \mu_0,\ (\Phi_1^w)_\# \mu_0\big) \leq \exp(C_{\text{geo}} L) \left( \int_0^1 \mathbb{E}_{X \sim \mu_t^v}\big[\|v(t, X) - w(t, X)\|_{g_X}^2\big] \mathrm{d}t \right)^{1/2},$$

*where $C_{\text{geo}}$ depends only on $(\kappa, r_0)$.*

**Remark 3** (How geometry enters). *The constant $C_{\text{geo}}$ arises from comparison-geometry control of normal coordinates and Grönwall-type stability along flow trajectories on a compact manifold. Qualitatively, $C_{\text{geo}}$ can be chosen to increase with curvature magnitude $\kappa$ and to deteriorate as the injectivity radius lower bound $r_0$ decreases. Thus, curvature influences the bounds through constants rather than the statistical exponent once uniform Lipschitz regularity is imposed.*

**Corollary 1** (End-to-end bound with approximation, discretization, and misspecification). *Assume 1–7. Let $\hat{\mu}_n = (\Phi_{1,K}^{\hat{v}})_\# \mu_0$. Then for any $\delta \in (0,1)$, with probability at least $1 - \delta$,*

$$W_2(\hat{\mu}_n, \mu^*) \leq \underbrace{W_2\big((\Phi_1^{v^\dagger})_\# \mu_0,\ \mu^*\big)}_{\text{misspecification gap}}$$

$$+ \exp(C_{\text{geo}} L) \underbrace{\left( \int_0^1 \mathbb{E}_{X \sim \mu_t^\dagger}\big[\|\hat{v}(t, X) - v^\dagger(t, X)\|_{g_X}^2\big] \mathrm{d}t \right)^{1/2}}_{\text{regression error along reference path}} + 2\, \mathcal{E}_{\text{disc}}(K).$$

*Moreover, by Assumption 7 and Theorem 1,*

$$W_2(\hat{\mu}_n, \mu^*) \leq W_2\big((\Phi_1^{v^\dagger})_\# \mu_0, \mu^*\big) + C'\, n^{-\frac{\beta}{2\beta+d}} \left( \log \frac{n}{\delta} \right)^\gamma + C'\, \mathcal{E}_{\text{approx}}(\mathcal{F}_n) + C' \eta^{1/2} + 2\, \mathcal{E}_{\text{disc}}(K),$$

*where $C'$ depends on $(\kappa, r_0, B, L, p_{\min}, p_{\max}, C_{\text{ent}})$. In the realizable case $W_2((\Phi_1^{v^\dagger})_\# \mu_0, \mu^*) = 0$, this yields the stated end-to-end rate.*

**Summary.** The bound decomposes the final generative error into: (i) a misspecification term (zero under realizability), (ii) a regression/statistical term with conservative exponent $\beta/(2\beta + d)$ (up to logarithms), (iii) approximation error through $\mathcal{E}_{\text{approx}}(\mathcal{F}_n)$, and (iv) numerical discretization error through $\mathcal{E}_{\text{disc}}(K)$. The exponent is governed by the intrinsic dimension $d$ via nonparametric regression complexity; geometry enters through constants once uniform Lipschitz regularity holds.

## 4 NUMERICAL EXPERIMENTS

We conduct numerical experiments to probe the empirical sample-size scaling suggested by our end-to-end bound (Corollary 1). In the (approximately) realizable regime where the misspecification gap $W_2((\Phi_1^{v^\dagger})_\# \mu_0, \mu^*)$ is small and the discretization error $2\mathcal{E}_{\text{disc}}(K)$ is controlled, Corollary 1 implies that the dominant statistical component of the generative error is governed by the regression rate of Theorem 1. For $\beta = 1$, this yields a conservative scaling exponent

$$\alpha_{\text{th}} = \frac{\beta}{2\beta + d} = \frac{1}{2 + d},$$

up to logarithmic factors and additional approximation/optimization terms. The goal of these experiments is not to certify worst-case optimality, but to check whether the observed error curves exhibit (i) clear power-law decay in $n$, (ii) intrinsic-dimension dependence through $d$, and (iii) limited sensitivity of the *exponent* to curvature (consistent with geometry entering primarily through constants in the stability bound).

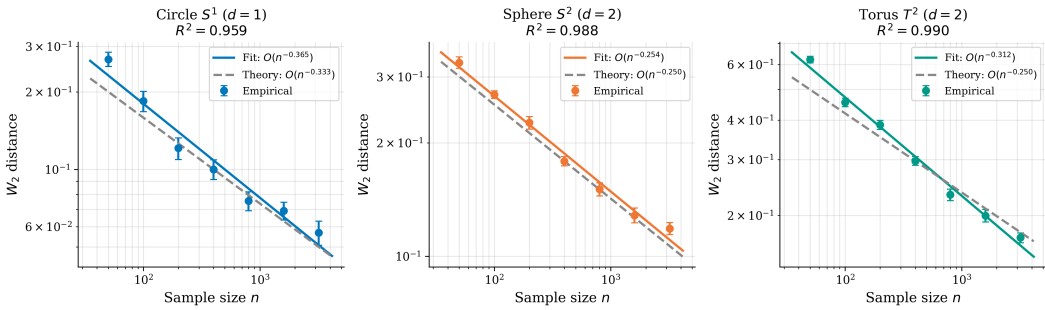

Figure 1: Convergence of empirical $W_2$ error as a function of sample size $n$ for three Riemannian manifolds. Points show means with 95% confidence intervals across trials; solid lines are fitted power laws; dashed lines indicate the conservative exponent $\alpha_{\text{th}} = \beta/(2\beta + d)$ with $\beta = 1$.

## 4.1 EXPERIMENTAL SETUP

**Manifolds.** We consider three compact Riemannian manifolds spanning different intrinsic dimensions and curvature regimes:

- **Circle** $S^1$ ($d = 1$): the unit circle embedded in $\mathbb{R}^2$.
- **Sphere** $S^2$ ($d = 2$): the unit sphere embedded in $\mathbb{R}^3$.
- **Flat torus** $T^2$ ($d = 2$): the product manifold $S^1 \times S^1$ with the flat metric.

**Target distributions.** On each manifold we use smooth target families (concentrated von Mises on $S^1$, von Mises–Fisher on $S^2$, and product von Mises on $T^2$). These choices are intended to yield stable CFM regression labels with bounded second moments (consistent with Assumption 2) and a well-behaved learned vector field in practice.

**Procedure and evaluation.** For each sample size $n \in \{50, 100, 200, 400, 800, 1600, 3200\}$, we:

1. draw $n$ i.i.d. samples from the target distribution $\mu^*$,
2. train a flow matching model using geodesic interpolation on the manifold,
3. generate $m = 500$ samples from the learned flow $\hat{\mu}_n$,
4. approximate the Wasserstein error using optimal transport with geodesic ground cost.

Concretely, we approximate $W_2(\hat{\mu}_n, \mu^*)$ by computing $W_2(\hat{\mu}_n^{(m)}, (\mu^*)^{(m)})$ between empirical measures constructed from $m$ samples from $\hat{\mu}_n$ and $m$ fresh samples from $\mu^*$. We repeat each experiment for 25 independent trials and report mean errors with 95% bootstrap confidence intervals. The convergence rate $\alpha$ is estimated via linear regression in log-log space by fitting $\log W_2 = c - \alpha \log n$.

**Evaluation error.** Because $W_2(\hat{\mu}_n, \mu^*)$ is estimated from empirical measures with a fixed sample size $m$, the reported values include an additional Monte Carlo/OT evaluation error that does not vanish with $n$ unless $m$ also increases. Consequently, fitted log-log slopes can be biased in finite samples; we interpret the fitted exponents as descriptive scaling in the tested regime rather than exact asymptotics.

## 4.2 RESULTS

**Power-law scaling and fitted exponents.** Table 1 summarizes the fitted exponents. Across all three manifolds, the log-log fits are strong ($R^2 > 0.95$), indicating clear power-law behavior over the tested range of $n$. For $S^2$ (where $d = 2$ and $\alpha_{\text{th}} = 1/4$), the empirical exponent $0.254$ is very close to the conservative prediction $0.250$, with a tight confidence interval. For $S^1$ (where $d = 1$ and $\alpha_{\text{th}} = 1/3$), the empirical exponent $0.365$ is moderately larger than $0.333$, which is consistent with

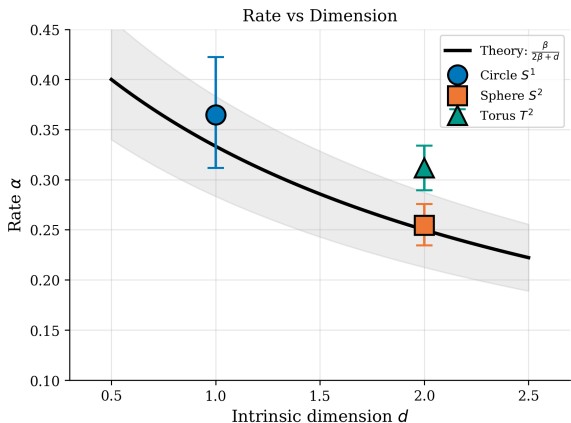

Figure 2: Fitted convergence exponent $\alpha$ versus intrinsic dimension $d$. The solid curve shows the conservative prediction $\alpha_{\text{th}} = \beta/(2\beta + d)$ with $\beta = 1$. Empirical exponents lie on or above the curve, consistent with conservative worst-case theory and finite-sample effects.

Table 1: Empirical scaling of flow matching error on Riemannian manifolds. The theory (Corollary 1) predicts a conservative exponent $\alpha_{\text{th}} = \beta/(2\beta + d)$ for the regression-driven component (here $\beta = 1$). The circle and sphere closely match this scaling, while the torus exhibits a faster slope in this finite-sample regime.

| Manifold | $d$ | Theoretical | Empirical | 95% CI | CI Width | $R^2$ |
|---|---|---|---|---|---|---|
| $S^1$ | 1 | 0.333 | 0.365 | [0.312, 0.422] | 0.111 | 0.959 |
| $S^2$ | 2 | 0.250 | 0.254 | [0.235, 0.276] | 0.041 | 0.988 |
| $T^2$ | 2 | 0.250 | 0.312 | [0.289, 0.334] | 0.045 | 0.990 |

Corollary 1 being an *upper bound*: it does not preclude faster decay for benign targets, favorable constants, or regimes where approximation/optimization effects are small.

For the flat torus $T^2$, we observe a noticeably larger exponent (0.312) than the conservative prediction 0.250. This does not contradict the theory. First, the product von Mises targets on $T^2$ have additional structure beyond generic Lipschitz densities, which can lead to better-than-worst-case behavior. Second, our reported Wasserstein errors are computed from finite samples ($m = 500$) using a numerical OT routine, which introduces an evaluation component that can affect fitted slopes. Finally, Corollary 1 includes approximation, optimization, discretization, and misspecification terms; when these are small relative to the statistical term, faster empirical decay can be observed.

**Dimension dependence.** A key qualitative prediction of our theory is that the *exponent* degrades with increasing intrinsic dimension $d$ through $\alpha_{\text{th}} = \beta/(2\beta + d)$, independent of the ambient embedding dimension. Figure 1 shows the log-log error curves and fitted slopes: the $d = 1$ circle decays faster than the $d = 2$ manifolds, consistent with intrinsic-dimension dependence. Figure 2 summarizes fitted exponents versus $d$ and overlays the curve $\alpha_{\text{th}} = 1/(2 + d)$ for $\beta = 1$. All empirical points lie on or above this conservative curve, consistent with the fact that Corollary 1 provides an upper bound and does not rule out faster finite-sample scaling.

**Curvature effects.** The stability inequality in Corollary 1 suggests that, once uniform Lipschitz regularity is imposed, geometry affects constants (via $C_{\text{geo}}$) rather than the statistical exponent. The two $d = 2$ manifolds ($S^2$ and $T^2$) show qualitatively similar power-law behavior in Figure 1. The difference in fitted exponents is plausibly attributable to target structure and finite-sample evaluation; isolating pure curvature-dependent constants would require a controlled study that varies curvature while holding the target family, training configuration, and OT evaluation accuracy fixed.

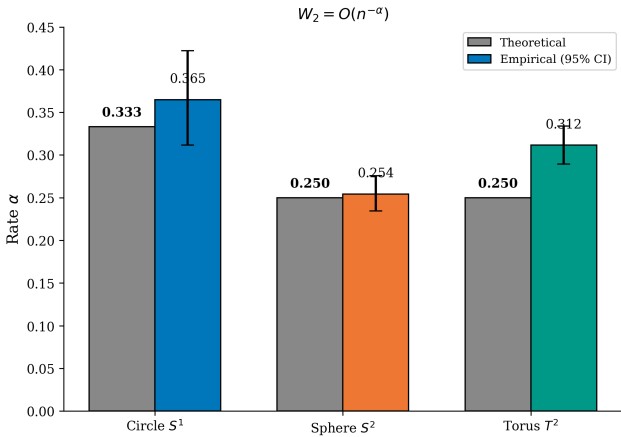

Figure 3: Comparison of the conservative theoretical exponent (gray) and fitted empirical exponents (colored) with 95% confidence intervals. The theory provides an upper-bound scaling $\alpha_{\text{th}} = \beta/(2\beta + d)$; empirical exponents meet or exceed this prediction in the tested regimes.

### 4.3 DISCUSSION

Overall, these experiments support three takeaways consistent with Corollary 1: (i) the observed error decays as a power law in $n$ (high $R^2$ fits), (ii) the exponent depends strongly on intrinsic dimension ($d = 1$ decays faster than $d = 2$), and (iii) curvature does not induce a qualitative change in the exponent in these settings. At the same time, the faster slopes observed on $T^2$ highlight that our guarantees are conservative worst-case upper bounds and can be pessimistic for structured target families and under finite-sample evaluation of $W_2$.

### 5 CONCLUSION

We analyzed the standard Riemannian Flow Matching pipeline on compact manifolds, learning a time-dependent vector field from conditional-flow-matching regression samples and generating it by numerically integrating the induced ODE. Our main result is an explicit end-to-end bound in $W_2$ that decomposes generative error into regression/statistical, approximation/optimization, and solver discretization contributions, plus a misspecification gap. The statistical exponent is controlled by intrinsic dimension through metric-entropy regression rates, while curvature and injectivity radius enter only through constants in a curvature-stable flow-to-measure stability inequality. Experiments on $S^1$, $S^2$, and $T^2$ exhibit robust power-law scaling and the expected degradation with dimension, supporting the conservative theoretical prediction. Future work includes relaxing the path-consistency/well-specified assumption, sharpening geometry-dependent constants, and extending the analysis beyond compact settings.

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

## A   APPENDIX

## B   PROOF OF THEOREM 1

We write the generic regression sample as $Z = (T, X_T, Y)$ and let $P$ denote its law; $P_n$ is the empirical measure of $\{Z_i\}_{i=1}^n$. For any measurable vector field $v : [0, 1] \times \mathcal{M} \to T\mathcal{M}$, define the squared loss

$$\ell_v(Z) := \|v(T, X_T) - Y\|_{g_{X_T}}^2, \qquad \mathcal{R}(v) = P\ell_v, \qquad \hat{\mathcal{R}}_n(v) = P_n\ell_v.$$

Recall the regression target $v^\dagger(t, x) = \mathbb{E}[Y \mid T = t, X_T = x]$.

**Envelope bound (used only for concentration).**   Because $\mathcal{M}$ is compact and the CFM label rule uses a bounded schedule (Assumption 2), the labels are uniformly bounded in norm; moreover, $\mathcal{F}_n$ is assumed to have finite $\|\cdot\|_\infty$-entropy. For the concentration steps below, we fix an envelope constant $B_0 < \infty$ such that

$$\|Y\|_{g_{X_T}} \le B_0 \text{ a.s.}, \qquad \sup_{v \in \mathcal{F}_n} \|v\|_\infty \le B_0, \tag{4}$$

So, in particular, $\|v^\dagger\|_\infty \le B_0$ and $\|v - v^\dagger\|_\infty \le 2B_0$ apply for all $v \in \mathcal{F}_n$. (All constants below may depend on $B_0$; this dependence is absorbed into $C_{\text{reg}}$ in the theorem statement.)

### B.1   STEP 1: EXCESS RISK IDENTITY FOR SQUARED LOSS

**Lemma 1** (Squared-loss regression identity). *For every measurable $v$,*

$$\mathcal{R}(v) - \mathcal{R}(v^\dagger) = \mathbb{E}\|v(T, X_T) - v^\dagger(T, X_T)\|_{g_{X_T}}^2.$$

*Equivalently, if we define the* excess loss

$$\Delta_v(Z) := \ell_v(Z) - \ell_{v^\dagger}(Z),$$

*then $P\Delta_v = \|v - v^\dagger\|_P^2$, where $\|v\|_P^2 := \mathbb{E}\|v(T, X_T)\|_{g_{X_T}}^2$.*

*Proof.* Expand $\ell_v = \|v - Y\|^2 = \|v - v^\dagger + v^\dagger - Y\|^2$:

$$\mathcal{R}(v) = \mathbb{E}\|v - v^\dagger\|^2 + \mathbb{E}\|v^\dagger - Y\|^2 + 2\mathbb{E}\left\langle v - v^\dagger, v^\dagger - Y \right\rangle.$$

Conditioning on $(T, X_T)$ and using $\mathbb{E}[Y \mid T, X_T] = v^\dagger(T, X_T)$ gives $\mathbb{E}[\langle v - v^\dagger, v^\dagger - Y \rangle \mid T, X_T] = \langle v - v^\dagger, v^\dagger - \mathbb{E}[Y \mid T, X_T] \rangle = 0$. Thus $\mathcal{R}(v) = \mathcal{R}(v^\dagger) + \mathbb{E}\|v - v^\dagger\|^2$, proving the claim. $\qquad\qquad\square$

By Lemma 1, bounding the $L_2(P)$ regression error reduces to bounding the excess risk.

### B.2   STEP 2: A LOCALIZED UNIFORM DEVIATION BOUND

Define the excess-risk functional $\mathcal{E}(v) := \mathcal{R}(v) - \mathcal{R}(v^\dagger) = P\Delta_v = \|v - v^\dagger\|_P^2$ and its empirical counterpart $\hat{\mathcal{E}}_n(v) := \hat{\mathcal{R}}_n(v) - \hat{\mathcal{R}}_n(v^\dagger) = P_n\Delta_v$ (note that $v^\dagger$ is used only for analysis).

A key property of squared loss under the envelope equation 4 is a Bernstein-type variance control:

$$\text{Var}(\Delta_v(Z)) \le \mathbb{E}[\Delta_v(Z)^2] \le C_1 B_0^2 \mathcal{E}(v) \qquad \text{for all } v \in \mathcal{F}_n, \tag{5}$$

and also $|\Delta_v(Z)| \le C_2 B_0^2$ almost surely. (This follows by expanding $\Delta_v = \|v - v^\dagger\|^2 + 2\langle v - v^\dagger, v^\dagger - Y \rangle$ and using $\|v - v^\dagger\|_\infty, \|v^\dagger - Y\|_\infty \le 2B_0$.)

We now state a standard peeling-and-net lemma tailored to our setting.

**Lemma 2** (Uniform comparison of empirical and population excess risk). *Let $p := d/\beta$ and assume Assumption 5. There exist constants $C_3, \gamma > 0$ (depending on $C_{\mathrm{ent}}$ and the polylog factor in Assumption 5) such that the following holds. For any $\delta \in (0,1)$, define*

$$r_n := C_3 \, n^{-\frac{1}{2+p}} \left( \log \frac{n}{\delta} \right)^{\gamma} = C_3 \, n^{-\frac{\beta}{2\beta+d}} \left( \log \frac{n}{\delta} \right)^{\gamma}. \tag{6}$$

*Then, with probability at least $1 - \delta$, simultaneously for all $v \in \mathcal{F}_n$,*

$$\mathcal{E}(v) \leq 2\hat{\mathcal{E}}_n(v) + C_3^2 r_n^2, \qquad \text{and} \qquad \hat{\mathcal{E}}_n(v) \leq 2\mathcal{E}(v) + C_3^2 r_n^2. \tag{7}$$

*Proof sketch (peeling + Bernstein + entropy).* We outline the standard argument and highlight the only place where the metric-entropy assumption is used.

*Step 1 (peeling in excess-risk shells).* For $j \geq 0$ define the shells

$$\mathcal{V}_j := \left\{ v \in \mathcal{F}_n : 2^j r_n^2 < \mathcal{E}(v) \leq 2^{j+1} r_n^2 \right\}, \qquad \mathcal{V}_{-1} := \{v \in \mathcal{F}_n : \mathcal{E}(v) \leq r_n^2\}.$$

It suffices to prove that, on an event of probability $\geq 1 - \delta$, for every $j \geq -1$,

$$\sup_{v \in \mathcal{V}_j} \left| \hat{\mathcal{E}}_n(v) - \mathcal{E}(v) \right| \leq \tfrac{1}{2} 2^{j+1} r_n^2 + C_3^2 r_n^2, \tag{8}$$

since then equation 7 follows by comparing any $v$ to the appropriate shell (the additive $C_3^2 r_n^2$ covers the small-radius region $\mathcal{V}_{-1}$ and the net-approximation error below).

*Step 2 (finite net + union bound).* Fix a shell $\mathcal{V}_j$ with $j \geq 0$ and set $s_j := 2^{(j+1)/2} r_n$ so that $\|v - v^\dagger\|_P \leq s_j$ for all $v \in \mathcal{V}_j$. Let $\epsilon_j := s_j/8$ and take an $\epsilon_j$-net $\mathcal{N}_j$ for $\mathcal{F}_n$ in the $L_2(P)$ metric $\|v - w\|_P := (\mathbb{E}\|v(T, X_T) - w(T, X_T)\|^2)^{1/2}$. Since $\|v - w\|_P \leq \|v - w\|_\infty$, the entropy assumption implies

$$\log |\mathcal{N}_j| \leq \log \mathcal{N}(\mathcal{F}_n, \epsilon_j, \|\cdot\|_\infty) \leq C_{\mathrm{ent}} \, \epsilon_j^{-p} \, \mathrm{polylog}(1/\epsilon_j). \tag{9}$$

*Step 3 (Bernstein on the net points).* For each $v \in \mathcal{N}_j$, Bernstein's inequality together with equation 5 yields

$$\mathbb{P}\left( \left| \hat{\mathcal{E}}_n(v) - \mathcal{E}(v) \right| > t \right) \leq 2 \exp\left( -\frac{c \, n \, t^2}{B_0^2 \, \mathcal{E}(v) + B_0^2 \, t} \right).$$

On shell $\mathcal{V}_j$, $\mathcal{E}(v) \asymp s_j^2 = 2^{j+1} r_n^2$. Choosing $t \asymp 2^{j+1} r_n^2$ and union-bounding over $\mathcal{N}_j$ using equation 9 shows that, provided $r_n$ is chosen as in equation 6, we have with probability at least $1 - \delta 2^{-j-2}$,

$$\sup_{v \in \mathcal{N}_j} \left| \hat{\mathcal{E}}_n(v) - \mathcal{E}(v) \right| \leq \tfrac{1}{4} 2^{j+1} r_n^2. \tag{10}$$

(The critical-radius choice equation 6 is exactly what makes the entropy term $\log |\mathcal{N}_j|$ negligible compared to $nr_n^2$ at $j = 0$; larger $j$ only decreases $\log |\mathcal{N}_j|$.)

*Step 4 (extend from the net to the full shell).* Take any $v \in \mathcal{V}_j$ and let $\pi_j(v) \in \mathcal{N}_j$ be a nearest net point, so $\|v - \pi_j(v)\|_P \leq \epsilon_j$. A direct expansion of $\Delta_v - \Delta_{\pi_j(v)}$ and Cauchy-Schwarz gives the deterministic bound

$$\left| (\hat{\mathcal{E}}_n - \mathcal{E})(v) - (\hat{\mathcal{E}}_n - \mathcal{E})(\pi_j(v)) \right| \leq C B_0 \|v - \pi_j(v)\|_P \left( \|v - v^\dagger\|_P + \|\pi_j(v) - v^\dagger\|_P \right) \leq C' \, 2^{j+1} r_n^2,$$

where we used $\|v - v^\dagger\|_P, \|\pi_j(v) - v^\dagger\|_P \leq s_j$ and $\|v - \pi_j(v)\|_P \leq s_j/8$. Absorbing constants and combining with equation 10 yields equation 8 on an event of probability $\geq 1 - \delta 2^{-j-2}$.

*Step 5 (sum over shells).* Taking a union bound over $j \geq 0$ and including $\mathcal{V}_{-1}$ (handled similarly with $t \asymp r_n^2$) gives equation 8 for all $j \geq -1$ with probability at least $1 - \delta$, which implies equation 7. $\square$

### B.3 STEP 3: CONCLUDE THE REGRESSION RATE FOR APPROXIMATE ERM

Let $v^\star \in \arg\min_{v \in \mathcal{F}_n} \mathcal{R}(v)$; equivalently, $v^\star$ minimizes $\mathcal{E}(v) = \|v - v^\dagger\|_P^2$ over $\mathcal{F}_n$. By definition, $\mathcal{E}(v^\star)^{1/2} = \mathcal{E}_{\mathrm{approx}}(\mathcal{F}_n)$.

Because $\hat{v}$ is an $\eta$-approximate ERM,

$$\hat{\mathcal{R}}_n(\hat{v}) \le \hat{\mathcal{R}}_n(v^\star) + \eta \quad \Longleftrightarrow \quad \hat{\mathcal{E}}_n(\hat{v}) \le \hat{\mathcal{E}}_n(v^\star) + \eta.$$

On the high-probability event of Lemma 2,

$$\mathcal{E}(\hat{v}) \ \le \ 2\hat{\mathcal{E}}_n(\hat{v}) + C_3^2 r_n^2 \ \le \ 2\hat{\mathcal{E}}_n(v^\star) + 2\eta + C_3^2 r_n^2 \ \le \ 4\mathcal{E}(v^\star) + (2C_3^2) r_n^2 + 2\eta.$$

Taking square-roots and using $\sqrt{a + b + c} \le \sqrt{a} + \sqrt{b} + \sqrt{c}$ yields

$$\|\hat{v} - v^\dagger\|_P \ \le \ \sqrt{2}\, C_3\, r_n \ + \ 2\, \mathcal{E}_{\text{approx}}(\mathcal{F}_n) \ + \ \sqrt{2\eta}.$$

Finally, substitute the definition of $r_n$ in equation 6 and absorb absolute constants into $C_{\text{reg}}$, proving Theorem 1. $\qquad\square$

*Proof of Theorem 2.* Fix $v, w$ satisfying the common bounds $(B, L)$ and let $\mu_t^v := (\Phi_t^v)_{\#}\mu_0$, $\mu_t^w := (\Phi_t^w)_{\#}\mu_0$. Let $X_0 \sim \mu_0$ and define the (a.s. unique) ODE solutions

$$X_t := \Phi_t^v(X_0), \qquad \widetilde{X}_t := \Phi_t^w(X_0), \qquad t \in [0, 1].$$

Consider the coupling $\pi := (X_1, \widetilde{X}_1)_{\#}\mu_0 \in \Pi(\mu_1^v, \mu_1^w)$. By definition of $W_2$,

$$W_2(\mu_1^v, \mu_1^w)^2 \ \le \ \mathbb{E}\big[d_{\mathcal{M}}(X_1, \widetilde{X}_1)^2\big]. \tag{11}$$

Thus it suffices to control the trajectory separation

$$r(t) := d_{\mathcal{M}}(X_t, \widetilde{X}_t), \qquad t \in [0, 1].$$

**Step 1: a first-variation inequality for the distance.** Both $t \mapsto X_t$ and $t \mapsto \widetilde{X}_t$ are absolutely continuous, hence $r(\cdot)$ is absolutely continuous as well. For almost every $t$ such that $\widetilde{X}_t$ is not in the cut locus of $X_t$ (and therefore the minimizing geodesic from $X_t$ to $\widetilde{X}_t$ is unique), the first variation formula yields

$$\frac{\mathrm{d}}{\mathrm{d}t}r(t) \ \le \ \big\|v(t, X_t) - \mathrm{PT}_{\widetilde{X}_t \to X_t}\, w(t, \widetilde{X}_t)\big\|_{g_{X_t}}, \tag{12}$$

where $\mathrm{PT}_{\widetilde{X}_t \to X_t}$ denotes parallel transport along the unique minimizing geodesic from $\widetilde{X}_t$ to $X_t$. (At the exceptional times when $\widetilde{X}_t$ lies in the cut locus of $X_t$, one may interpret equation 12 with the upper Dini derivative; since this occurs on a set of Lebesgue measure zero for absolutely continuous curves, integrating the inequality below is unaffected.)

**Step 2: reduce to a Grönwall inequality using spatial Lipschitz regularity.** Add and subtract $w(t, X_t)$ in equation 12 to obtain, for a.e. $t$,

$$\frac{\mathrm{d}}{\mathrm{d}t}r(t) \le \|v(t, X_t) - w(t, X_t)\|_{g_{X_t}} + \big\|w(t, X_t) - \mathrm{PT}_{\widetilde{X}_t \to X_t}\, w(t, \widetilde{X}_t)\big\|_{g_{X_t}}. \tag{13}$$

To control the second term we use the spatial Lipschitz assumption in an intrinsic form. For each $t$, define the parallel-transport Lipschitz seminorm of a vector field $u(t, \cdot)$ by

$$\|u(t, \cdot)\|_{\mathrm{Lip}, \mathrm{PT}} := \sup_{x \ne y} \frac{\|u(t, x) - \mathrm{PT}_{y \to x} u(t, y)\|_{g_x}}{d_{\mathcal{M}}(x, y)},$$

where $\mathrm{PT}_{y \to x}$ is parallel transport along a minimizing geodesic from $y$ to $x$. On a compact manifold with bounded geometry (Assumption 1), any chart-based Lipschitz bound is equivalent to $\mathrm{Lip}, \mathrm{PT}$ up to a constant depending only on $(\kappa, r_0)$; in particular there exists $C_{\text{geo}} = C_{\text{geo}}(\kappa, r_0) \ge 1$ such that, for all $t \in [0, 1]$ and all $x, y \in \mathcal{M}$,

$$\big\|w(t, x) - \mathrm{PT}_{y \to x} w(t, y)\big\|_{g_x} \ \le \ C_{\text{geo}} \|w(t, \cdot)\|_{\mathrm{Lip}}\, d_{\mathcal{M}}(x, y) \ \le \ C_{\text{geo}} L\, d_{\mathcal{M}}(x, y). \tag{14}$$

Applying equation 14 with $(x, y) = (X_t, \widetilde{X}_t)$ in equation 13 yields

$$\frac{\mathrm{d}}{\mathrm{d}t}r(t) \ \le \ a(t) + C_{\text{geo}} L\, r(t), \qquad a(t) := \|v(t, X_t) - w(t, X_t)\|_{g_{X_t}}. \tag{15}$$

Since $X_0 = \widetilde{X}_0$ under our coupling, we have $r(0) = 0$. Grönwall's inequality applied to equation 15 gives

$$r(1) \ \le \ \int_0^1 e^{C_{\text{geo}} L(1-s)}\, a(s)\, \mathrm{d}s. \tag{16}$$

**Step 3: square, take expectations, and identify the trajectory norm.** By Cauchy–Schwarz and the crude bound $\int_0^1 e^{2C_{\text{geo}}L(1-s)}\mathrm{d}s \leq e^{2C_{\text{geo}}L}$,

$$r(1)^2 \;\leq\; \left(\int_0^1 e^{2C_{\text{geo}}L(1-s)}\mathrm{d}s\right)\left(\int_0^1 a(s)^2\mathrm{d}s\right) \;\leq\; e^{2C_{\text{geo}}L}\int_0^1 \|v(s,X_s) - w(s,X_s)\|_{g_{X_s}}^2 \,\mathrm{d}s. \tag{17}$$

Taking expectation over $X_0 \sim \mu_0$ and using that $X_s \sim \mu_s^v$ for each $s$,

$$\mathbb{E}[r(1)^2] \;\leq\; e^{2C_{\text{geo}}L}\int_0^1 \mathbb{E}_{X\sim\mu_s^v}\big[\|v(s,X) - w(s,X)\|_{g_X}^2\big]\,\mathrm{d}s. \tag{18}$$

Finally, combining equation 11 and equation 18 and taking square-roots yields

$$W_2(\mu_1^v,\mu_1^w) \;\leq\; e^{C_{\text{geo}}L}\left(\int_0^1 \mathbb{E}_{X\sim\mu_s^v}\big[\|v(s,X) - w(s,X)\|_{g_X}^2\big]\,\mathrm{d}s\right)^{1/2},$$

which is the claimed stability inequality. $\qquad\square$

