# OpenReview forum: "Geometry-Grounded Flow Matching on Compact Manifolds"
_ICLR.cc/2026/Workshop/GRaM — ICLR 2026 Workshop GRaM Poster_

### Official Review · Reviewer_tMpj · 2026-02-08

**Rating:** 7
**Confidence:** 2

**Review:**

The authors study the theoretical limits of Riemannian flow matching.
Their analysis considers the statistical error, the approximation/optimization error, and the discretization error in the problem.
They show that their stability theorem is robust to geometry (curvature, injectivity radius) when Lipschitz regularity exists, and the learning rate is only a function of the intrinsic dimension.
The paper is concluded with experiments.

pros:
- relevant to the scope of the workshop
- interesting theoretical contributions

cons:
- the connection to prior work is not very detailed in the paper

This is an interesting theoretical paper on the interaction between geometry and flow matching. I suggest the authors provide a more detailed comparison with prior work on this topic to better position their paper

Minor: Where is "Appendix A" it looks like it is either missing or there is a typo there

**Pmlr Suitability:**

Yes

---

### Official Review · Reviewer_JXbd · 2026-02-24
**Strong characterization of Riemannian flow matching, but difficult to follow at times.**

**Rating:** 6
**Confidence:** 3

**Review:**

# Summary:

The paper provides bounds on the Wasserstein 2 distance between a probability distribution defined on a Riemannian manifold and the generative distribution of a learned Riemannian flow matching model in terms the number of samples used for training, complexity and regularity of the approximation class, optimization error, the geometry of the manifold, and solver error incurred by numerical simulation of the learned ODE.  The main theoretical contribution is that, under all of the assumptions stated in the paper, the regression error for the flow matching objective that bounds the W2 distance includes a constant multiplier that depends on the sectional curvature and injectivity radius of the manifold. The experiments validate the bound on small scale experiments.

# Strengths:
- The paper gives a comprehensive characterization of learning using Riemannian flow matching, which gives offers useful insights into how and when Riemannian flow matching can be more challenging than Euclidean flow matching.
- The theory accounts for all of the sources of error that appear in practice.
- All of the assumptions are explicitly laid out in the paper.

# Weaknesses:
- The paper is hard to follow in section 3 at times, primarily because there is no exposition and just statement of results with a few remarks that are difficult to parse.  For example, remark 3 was particularly dense and too low level to help interpret $C_{\text{geo}}$ without needing to look at the proof in the appendix.
- Parts of the paper are stated without explanation.  For example, I'm not entirely sure why choosing $\beta=1$ is a reasonable choice for the experiments.

# Minor:
- The letter K is overloaded as the sectional curvature in Assumption 1, number of solver steps throughout the paper, and number of basis functions in Remark 1.
- The definition of approximation error I think is supposed to appear in section 3.3 instead of where it is above Remark 1 because it it not referenced until then.

**Pmlr Suitability:**

Yes

---

### Official Review · Reviewer_EauJ · 2026-02-24

**Rating:** 6
**Confidence:** 3

**Review:**

## Summary

This paper analyzes Riemannian Flow Matching (RFM) for generative modeling on compact Riemannian manifolds. It formalizes the standard pipeline—construct conditional-flow-matching regression tuples, learn a time-dependent vector field with squared-loss regression, and sample by integrating the resulting ODE—and derives an end-to-end upper bound on distributional error (Wasserstein-style on the manifold). The bound decomposes into three interpretable contributors: (1) statistical error from estimating the conditional-mean velocity, (2) approximation/optimization error from the function class and learning procedure, and (3) numerical discretization error from the ODE solver. The paper also provides a stability inequality connecting vector-field error to measure error on manifolds, with geometry entering through constants under boundedness/Lipschitz assumptions. Toy experiments on the circle, sphere, and torus show power-law scaling with sample size consistent with intrinsic-dimension dependence.

## Strengths

- Clear error decomposition. Separating learning, misspecification/optimization, and solver discretization is genuinely helpful for understanding how and why RFM can fail.-
Geometry-aware stability viewpoint. The flow-to-distribution stability step is a useful conceptual contribution, moving beyond “Euclidean intuition” for error accumulation.
- Intrinsic-dimension emphasis. The analysis aligns with the motivation for manifold modeling and communicates the key scaling story cleanly.
- Experiments support the narrative. The empirical scaling trends broadly match the theoretical message.

## Weaknesses

Assumptions are strong and may blunt practical value. Conditions like global boundedness of densities, uniform Lipschitz control, compactness, and especially well-specified/path-consistent constructions are not easy to guarantee in realistic setups. When violated, the bound may be loose or uninformative.
- Geometry hidden in constants could still matter a lot. Saying curvature/injectivity radius “only affect constants” can mask regimes where those constants become huge, limiting guidance for practitioners.
- Weak linkage to modern neural parameterizations. The hypothesis-class/metric-entropy framing is generic; it’s not obvious how it translates to the architectures and regularization actually used for manifold vector fields.
- Empirical scope is narrow. Low-dimensional toy manifolds and scaling plots are a good sanity check but don’t establish competitiveness, robustness, or usefulness in harder settings. Comparisons and ablations (solver steps, schedules, Lipschitz control) are limited.

**Pmlr Suitability:**

Yes

---

### Meta-Review · Area_Chair_dxDf · 2026-02-25

**Decision:**

Accept

**Metareview:**

The reviewers agree that the paper provides **solid theoretical guarantees** for the Riemannian flow matching model on compact manifolds. We are happy to accept this paper to our proceedings.

We strongly encourage the authors to improve their paper for the camera-ready version. In particular: differentiating between the K (curvature, steps, number of basis functions) and adding more explanation when introducing the variables in section 3. The appendix sections can be a bit clearer too, as "Appendix A" is empty, "Theorem 1" is its own section, while "Theorem 2" is within the text (and so the proof seems harder to find).

**Relevance To Proceedings:**

Yes — suitable for PMLR (long paper)

**Relevance To Workshop:**

Yes — suitable for GRaM

---

### Decision · Program_Chairs · 2026-03-02

Accept (Poster)